# Facing Conspiracies: Biden's Counter-Speech to Trumpist Messages in the Framework of the 2020 US Elections

Concha Pérez-Curiel [ID], Rubén Rivas-de-Roca *[ID] and Ricardo Domínguez-García [ID]

Department of Journalism II, University of Seville, Avda. Américo Vespucio, s/n. 41092 Seville, Spain
* Correspondence: rrivasderoca@us.es

**Abstract:** The climate of division and polarization in the US politics is increasing, going beyond the time in the office of a specific leader. Several political or technological challenges have ended up eroding this trust, making social cohesion difficult. In this context, this research examines the communication strategies of the elected president Biden after the 2020 elections, shedding light on how his legitimacy was built. All the messages that the Democrat published on his personal Twitter account (@JoeBiden) were collected, from the day after the presidential elections (4 November 2020) until his inauguration as president of the United States (20 January 2021). Using a content analysis method on issue/game frame and dissemination of the messages ($n = 379$), and an analysis of the 100 first keywords, results showed a plan of the Democratic candidate to reinforce the role of public institutions but without interaction with the polarized electorate. In this sense, the strategies of the president-elect related to the promotion of political action, the call for unity, and the fight against the pandemic stood out. The frequent use of words with a positive attitude reveals how Biden avoided confrontation with Donald Trump.

**Keywords:** conspiracy; illegitimacy; Biden; Trumpism; Twitter; US elections





## 1. Introduction

The episodes of revolutionary activism against the US Capitol in 2021, triggered by the action of Donald Trump on social media, represent the last moment of a presidency marked by the attempt to spread theories of fraud and illegitimacy of the voting processes. The image of the assault was powerful and caught the attention of world public opinion, illustrating the crisis of democracies [1]. This event was proof of the influence of Donald Trump's speech on voters, applying a discursive strategy based on falsehood disseminated on Twitter [2].

According to reports, the assault on the US Capitol fostered feelings of fear and anxiety among citizens [3]. However, the fragmentation of the public is much more worrying in the current political context. Statistical analysis of benchmark public polls [4] revealed how domestic Republican criticism of the former president decreased in the months after the riots and how the facts of the assault were even questioned. Hence, the political ideology impacted citizenship, using lies as electoral strategies [5] that led to a post-truth era of ignorance [6].

Although Trump was criticized by some of the public throughout the 2020 campaign, the legacy of his messages is still present. The xenophobic statements against minorities [7], the disruptive domestic policy of the past and grandiloquent foreign policy [8], and the employment of spectacularization [9] and cyberpopulism [10,11] are features of his discourse. These trends affect the Biden government, reaching high levels of polarization. As the campaign was marked by polarized statements, it is noteworthy to know how Biden approached the post-election period on Twitter.

A distrust in the management of public affairs exists together with strongly ideological opposing positions and messages about the illegitimacy of the elections. As long as polarization seems to be a key factor in the shaping of the current US democracy, Biden

could develop a consensus approach centered on the idea that "America is back" as collected by specialized reports [12] that asked for a pact between institutions to respond to citizens' mistrust. The literature has consistently described the information disorder and the degree of misinformation, which alerted public institutions [13] and social platforms to intervene [14], sanctioning the electoral use of leaders and fraudulent content of their messages. Nonetheless, it would be relevant to illustrate how political competitors react to disinformation, especially if they develop a counteractive communication. This sort of research could contribute to ongoing works in the battle against distrust in politics.

Bearing this context in mind, we content-analyzed the issues, strategies, and dissemination of messages shared by the president-elect after the presidential elections (4 November 2020). Moreover, we studied the first 100 keywords used by Biden, which allowed assessment of his differential values compared to the politics and communicative management of Donald Trump. Lastly, an open debate is addressed on the role and influence of political leaders in electoral stages and the projection of their proposal in time of governance.

In the context of a struggle to delegitimize the result of the 2020 US elections, which came to the fore in the assault on the Capitol in Washington DC and the crisis of Western democracies the objective of this research was to examine the strategies that allowed Joe Biden to defend the American democratic system against the Trumpist criticism of electoral theft. From this approach, the following research questions are posed:

RQ1. How did Joe Biden use Twitter to counteract the narrative of electoral delegitimization?

RQ2. What were the main issues and strategies of Biden's speech on Twitter?

RQ3. Were the keywords used by Biden a rhetorical device to defend the legitimacy of the democratic process and his electoral victory?

## 2. Theoretical Framework

### 2.1. The State of Political Legitimacy and Social Cohesion

The elections in Western democracies are currently shaped by the confluence of factors such as populism, disinformation, and propaganda [15]. This indicates a crisis in the trust of institutions. Specifically, Trump was a candidate and a president that was able to obtain international attention thanks to his aggressive cyber rhetoric. In the two elections in which he led the Republican Party, an attack on globalization and integration policies, and the establishment, emerged. His speech also referred to supranational entities and groups such as immigrants or refugees from a critical perspective [16]; meanwhile, he confronted the media, especially regarding the international field. For Trump, Twitter was the main network used to disseminate messages [17].

The divergence of opinions on the results of the elections appears around a story of conspiracy and illegitimacy that prevails on social media. The consequence was the projection of these conflicting positions in a visibly divided electorate [18]. Data show that as many as 70% of Republican voters described Joe Biden's victory as fraud. Donald Trump's campaign reached its peak of propaganda with an assault on US Capitol, promoting disaffection institutions [19].

The previous literature developed several studies that explored the personality of the Republican president from several approaches. For instance, his role as a digital influencer [20], the connection with extreme right-wing populism [21], the narratives of his political discourse [22], and his control of marketing were addressed. The purpose of those studies was to further our understanding of how Trump could mobilize voters.

Conversely, the figure of Joe Biden has been little explored. In this sense, there are more studies on his management of the presidency rather than partisan views. The measures of his presidency were analyzed with regard to coronavirus [23], climate change [24], economic policy [25], and foreign policy [26]. Furthermore, the transatlantic cooperation alliance with European countries, the return to some international institutions, and the challenging influences of China or Russia have determined his agenda. In comparative

terms with Trump, both leaders were studied from the perspective of discourse rhetoric [27] or economic studies [28].

*2.2. Polarization and Distrust in the US Society*

Prior scholarship on the leadership and influence of Biden [29,30] revealed a style of politics that is quite different from Donald Trump's communication actions on social media [31]. Therefore, it is relevant to contribute to the topic on how Biden responds to these challenging messages, struggling with fraud and conspiracy. It should be noted that this took place within an anti-press rhetoric boosted by Trump [32], which shows how the media culture and political participation are in transition.

Recent events such as the Supreme Court abortion decision illustrate the conservative turn of the US society, but the public presence of strong conservative positions started with the action of the Tea Party against Obama [33]. As a consequence, the perceptions of the state of the Union remain negative. Although Joe Biden was positively evaluated when he was elected, his approval rating has decreased, and many do not trust the president [34].

Discussions on the role of information and political knowledge in the US [35] have been replaced by the analysis of the role of disinformation, moving from informed to disinformed citizens [36]. The power of false messages is emphasized, as they take advantage of the possibilities of social media platforms. Furthermore, the COVID-19 pandemic altered the flows of communication among the actors of political communication, i.e., politicians, journalists, and citizens [37].

The high level of distrust lies in the fact that populist leaders benefit from social media [38] while the fragmentation of the US society is another reason. On this matter, the US is a conflictive democracy in terms of social unity. Several political conflicts are identified by the majority of Americans. It is true that other countries suffer from these problems, but the Pew Research Center survey highlighted that the US is much more divided.

As stated, the polarization in the American political system has long-lasting roots. Many observers of US politics have identified that partisan polarization is shaping the work of the Congress. There is a crisis of liberalism [39] that explains why the ideological differences between Democrats and Republicans have been expanded in recent years. The rise of populism in Europe and America is a shared phenomenon that connects with the worldwide spread of disinformation [40].

Populism has triggered several risks for representative democracy, since this movement works as a thin-centered ideology that manages a catchy communication style [41]. The appearance of more authoritarian regimes overlaps with this preference for populist leaders. This also affects consolidated democracies such as the US since many citizens do not consider democracy as the most desirable system. There is a pattern of weakness and loss of importance of parliamentarism [42,43], but traditional parties still play a role. Thus, it is interesting to delve into the speeches of the leaders and how they promote or counteract conspiracy.

### 3. Materials and Methods

For this study, a content analysis method was conducted, both quantitative/qualitative [44,45] and discursive [46,47]. The analysis focused on the issue frame/game frame theory [48], which is very relevant to understanding whether political actors prefer issues or strategies in their communication. Specifically, Twitter was chosen as the social network to study, given its key position for political communication in electoral processes. Likewise, it was the main social platform used by Donald Trump in order to disseminate his messages. In 2016, the use of Twitter in the US elections broke all the established distinctions that observers had used to describe the campaigns, starting a new era of political discourse [49]. This makes it relevant to analyze this specific social network.

As for research design, with the purpose of knowing the story behind the winner of the 2020 US elections, the whole presidential transition process was analyzed. Hence, we collected all messages published by Biden from his personal Twitter account (@JoeBiden),

ranging from the day after the election (4 November 2020) to his inauguration as president of the country (20 January 2021).

Taking all of the above into account, the sample universe was made up of the 379 tweets that Joe Biden posted throughout the defined timeframe. Previous methodological models were considered [50,51], learning from them the importance of prioritizing quality over quantity in discourse analysis. On the basis of this approach, a coding manual was carried out between three coders to identify the defining features of Joe Biden's speech. A total of 10 quantitative, qualitative, and discursive variables were used. The statistical program used to process the data was IBM SPSS Statistics, Version 28, which allows descriptive findings to be obtained. Statistical tests were not suitable due to the small sample size. Furthermore, the reliability of the coders was calculated using Scott's pi formula, reaching an error level of 0.98 among the researchers.

First, our study was structured within three large frameworks: issue frame (theme), game frame (strategies), and dissemination of the message (hypertextuality and virality). To explore the agenda (issue frame) of Biden's tweets, a random pre-sampling of 50 tweets was applied to the general sample ($n1 = 379$) to determine the main thematic categories: electoral process, economy, health and COVID-19, science, social affairs, international relations, security, storming the Capitol, inauguration, and others [52]. The following codebook was employed to measure the issue frames (Table 1).

**Table 1.** Codebook used for the analysis of the agenda (%).

| Topics | Description |
| --- | --- |
| Electoral process | Tweets that allude to the elections. |
| Economy | Tweets concerning economic policy and its topics. |
| Health and COVID-19 | Tweets on health and the COVID-19 pandemic. |
| Science | Tweets that mention science as a way to move forward the society. |
| Social affairs | Tweets that refer to issues related to social policy. |
| International relations | Tweets on the political relationship with other areas of the world. |
| Security | Tweets that mention home affairs. |
| Storming the Capitol | Tweets that refer to the US Capitol attack. |
| Inauguration | Tweets on the start of Joe Biden in the US presidential office. |
| Other | Unclassifiable tweets in the previous categories. |

Regarding discursive strategies, a similar procedure was adopted by establishing the following categorization: defense of their victory, defense of the electoral process, mobilization in the partial elections in Georgia, nomination of members of the Government, announcement of government measures, request for funds, and others. Our study takes into account individual tweets and responses but not the retweets since they are not relevant to identifying Biden's agenda. However, Biden's narrative was also studied through the tone of his tweets based on previous research [53]. Three levels were created, positive, neutral, and negative, which allowed verification of the attitude of the Democrat leader in each message. For instance, on 4 January 2021, Biden posted a message with a positive tone "Georgia—If you haven't returned your absentee ballot yet, drop it off at a drop box in your county today. If you're voting in-person tomorrow, make your plan today. Let's flip the Senate". By contrast, the neutral or negative tweets were criticism of the conflictive situation: "We need to remember: We're at war with a virus—not with each other" (25 November 2020).

In addition, some variables related to the production process of the tweet were analyzed, with the aim of delving into the planning of these messages. Any added audiovisual content (only text, image, video, retransmissions, links, or quoted tweets) and the use of hashtags and quotes were studied, and their purpose (whether covering an act of Biden or if developed specifically for social networks).

With the aim of assessing the viralization and influence capacity of each of the tweets, and taking previous studies as a reference [54], a new formula was developed that assigns a double value to retweets versus likes and replies. This was due to the fact that Twitter gives

greater visibility to the retweet, showing the message on the timeline of the person who shares it. In the same vein, the viralization formula was based on the sum of likes, replies, and retweets multiplied by 2, divided by the number of published tweets (viralization capacity = (SUM retweets × 2 + SUM likes + SUM responses)/SUM published tweets). We used the viralization formula because of the need to determine the topics that promote higher engagement. These issues may be useful to reconnect with a distant audience.

Lastly, as also supported by previous research [55,56], an analysis of keywords was carried out. This was applied on a corpus of words (4638) extracted from the published tweets by Joe Biden. To manage the corpus, the AntConc program [57] was used, with which the terms were lemmatized [58,59], and a stop list was applied in order to exclude frequent words such as prepositions, conjunctions, determiners, or auxiliary verbs from the analysis. The most used keywords (50) were obtained together with how they were combined with each other, enabling us to observe the thematic and discursive marks that characterized Biden's storytelling on Twitter.

## 4. Results

### 4.1. Biden's Use of Twitter as President-Elect

First, we shed light on the general sample of tweets published by Joe Biden ($n1$ = 379) throughout the presidential transition process. Data show that the Democrat did not use Twitter in a massive way, posting an average of 4.86 messages a day. His volume of tweets revealed several correlations in terms of the days chosen for the most and the least dissemination of content.

On this matter, the dates of massive production of messages matched the milestones of the US electoral process: the days after the elections (from 4 to 7 November), the start of the presidential transition process (24 and 25 November), the proclamation of Biden as president-elect (14 and 15 December), the partial elections in Georgia (4 and 5 January), and the ratification of his victory by Congress, which occurred on the same day as the assault on the US Capitol (6 and 7 January).

By contrast, the Democrat posted less on Twitter or even did not post any tweets at all on some weekends (8 November, 14 and 15 November, 21 and 22 November, and 18 December) nor during Christmas (25 December) and the New Year (1 January). All these findings show a relationship between his presence on the social network and the own president's work schedule.

### 4.2. The Thematic and Strategic Agenda of Biden

The analysis of Joe Biden's tweets showed that his thematic agenda was plural, as several priority issues emerged. According to Table 2, most of the messages connected with the electoral process (32.3%), but there were other highly cited topics: the impact of the COVID-19 pandemic on the health system (24%) and economic or employment measures (11.3%). The inauguration of the presidency (7.1%), security and defense (6.6%), social affairs (6.1%), and the role of science and climate change (4.7%) were also key for him. Conversely, the scant presence (1.1%) of foreign affairs and his relationship with other countries was remarkable, considering the traditional international dimension of the Democrats.

Moreover, the time distribution of topics showed a trend toward a planning of communication (Figure 1). We found that the only issues that were continuously present in Biden's story were the electoral process and the fight against COVID-19 while the remaining issues were frequently grouped by days. Thus, Joe Biden usually thematized his messages for a day or two, matching the nomination of the members of his government in charge of an issue and the announcement of the main measures that he would promote in that field. Examples of this include the presentation of the security teams on 24 November, economy on 1 December, health on 7 December, and education on 23 December.

**Table 2.** Distribution of tweets based on issue frames (%).

| Issue Frames | Frequency |
|---|---|
| Electoral process | 32.3 |
| Health and COVID-19 | 24.0 |
| Economy and employment | 11.3 |
| Inauguration of the presidency | 7.1 |
| Security and defense | 6.6 |
| Social affairs | 6.1 |
| Science and climate change | 4.7 |
| Assault on US capitol | 3.2 |
| Foreign affairs | 1.1 |
| Other | 3.6 |

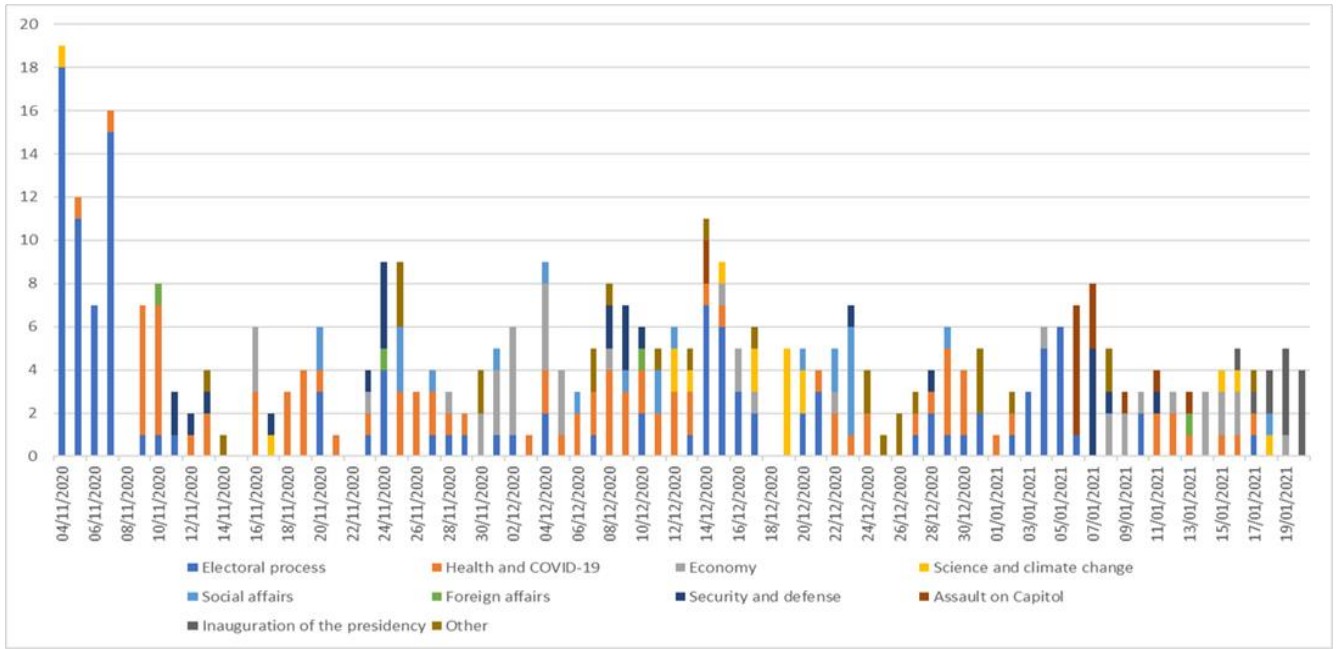

**Figure 1.** Use of issue frames per day.

With regard to audiences, the analysis of the virality of the tweets published by the Democrat (Figure 2) revealed that these messages were very well received by the public on Twitter compared to previous studies [60]. They received 24,378 retweets, 233,260 likes, and 16,113 replies, indicating an average viral ratio of 298,129. When considering the different topics addressed by Biden, a huge response from citizens was detected for political events of the US democracy, such as Biden's inauguration (with a value of 790,604, according to the formula developed for this research) and the assault on the Capitol (573,740). Furthermore, the dissemination of tweets related to the electoral process (365,911) stood out, followed by those on health (297,157).

In relation to this last section, it should be taken into account that the president-elect tended to publish messages on COVID-19, especially advising the use of a mask or safe distancing. By contrast, citizens showed a much lower interest in social affairs (131,399), security (150,795), and the economy (154,010). In this sense, citizens interacted to a large extent on issues that concerned the democratic process and were more reluctant to do so with thematic issues based on promises for the future.

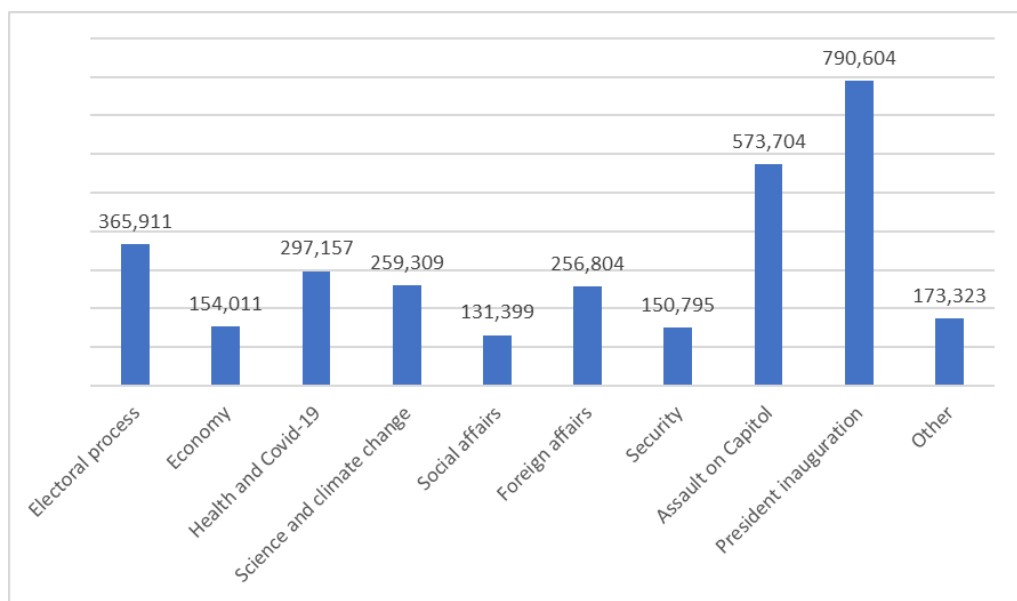

**Figure 2.** Viralization capacity according to issues.

Moreover, the tone in the Democrat's tweets showed a clear majority tendency (51.7%) for positive messages, defending a better future for the US, which overlapped with dealing with the pandemic and promoting better economic conditions. There was a very high percentage of neutral tweets (42.5%), especially those in which the president-elect limited himself to announcing members of his future government or took a position on specific issues. Conversely, there were a very small number of messages with a negative bias (5.8%), limited to those in which Biden criticized the management of the outgoing president, Donald Trump. In terms of the viralization rate, it was confirmed that the positive messages (331,232) and the negative ones (327,611) generated greater interest in the public than the neutral ones (253,802).

From the point of view of political strategies (Figure 3, the use that Joe Biden made of the announcement of measures (21.1%) that he would promote once he became President of the United States and of presentations by members of his future cabinet (12.4%) was striking. The Democrat established his legitimacy as the winner of the elections and started the presidential transition process, despite Donald Trump's obstacles. In addition, Biden frequently defended the electoral process (11.6%) and his own victory (10.8%).

However, it is noteworthy that, on very few occasions (4.7%), Biden came into direct confrontation with the Republican candidate and outgoing president, Donald Trump, who attempted to delegitimize the electoral results. There was also a high percentage of messages from Biden in which he was committed to other types of strategies, focusing on personalizing the candidate and promoting engagement with users. Examples of this are the congratulatory messages on different holidays or the tweets that he published urging people to wear a mask to deal with the pandemic.

*4.3. Shaping Biden's Messages on Twitter*

Our study also examined the employment of resources such as image or video in the tweets. The aim was to determine their production routines and how these messages were planned. Accordingly, a clear majority of messages observed did not include any audiovisual or hypertext element; hence, they were exclusively based on the text of the message (47%). Nevertheless, there were several messages with videos (21.1%) of the Democrat leader, and broadcasting of events (10.6%), images (8.2%), or links (8.2%). If we consider the virality of the tweets according to the presence of resources (Figure 4), we find that tweets that only used textual messages had the greatest impact (476,335), doubling those that incorporated images (197,710) or videos (182,209).

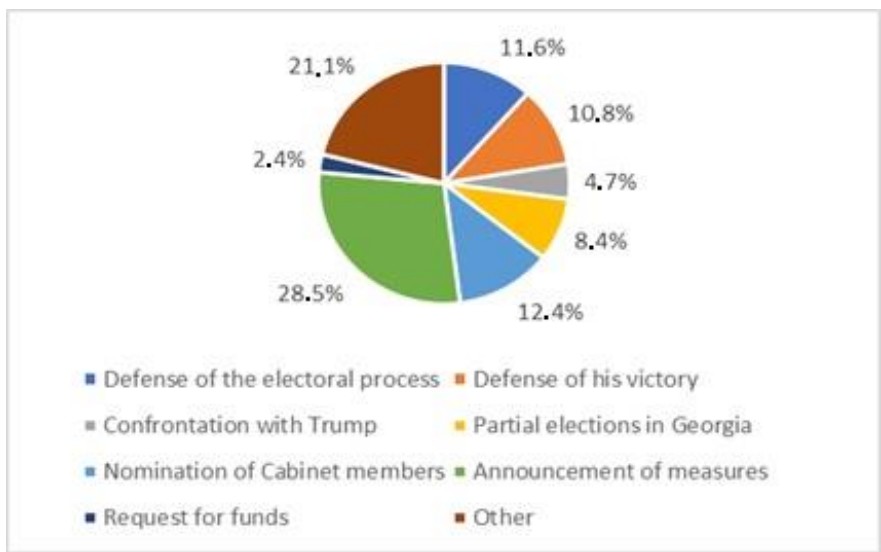

**Figure 3.** Distribution of tweets based on game frame.

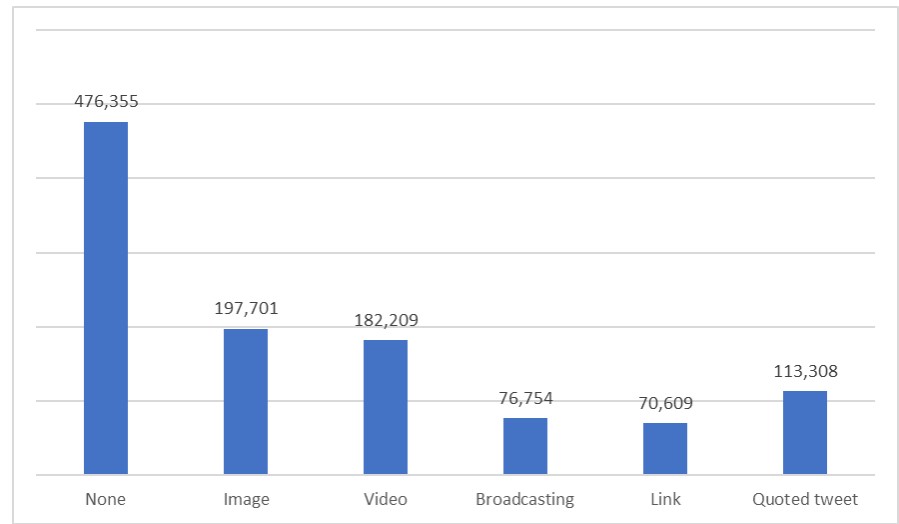

**Figure 4.** Impact of tweets according to the use of resources.

Regarding the use of quotes in tweets, a very low use was detected (9.5%) by Joe Biden throughout the presidential transition period. As for the accounts cited (Figure 5), the mentions of the elected vice president Kamala Harris (38.9%) stood out. Harris was followed by the Senate candidates in the Georgia elections Jon Ossof and Raphael Warnock (13.9%), the nominee for Secretary of Transportation Peter Buttigieg (8.3%), and the Biden Presidential Inauguration Committee (8.3%). On the other hand, only five tweets using hashtags (0.3% of the total) were found, and they referred to specific events (#inauguration2021, #MLKDay, #HappyDiwali, #SmallBusinessSaturday, and #HumanRightsDay). Lastly, it should be noted that the vast majority of the messages were created for social networks (71%) while a minority of tweets (29%) disseminated content on Biden's agenda of events.

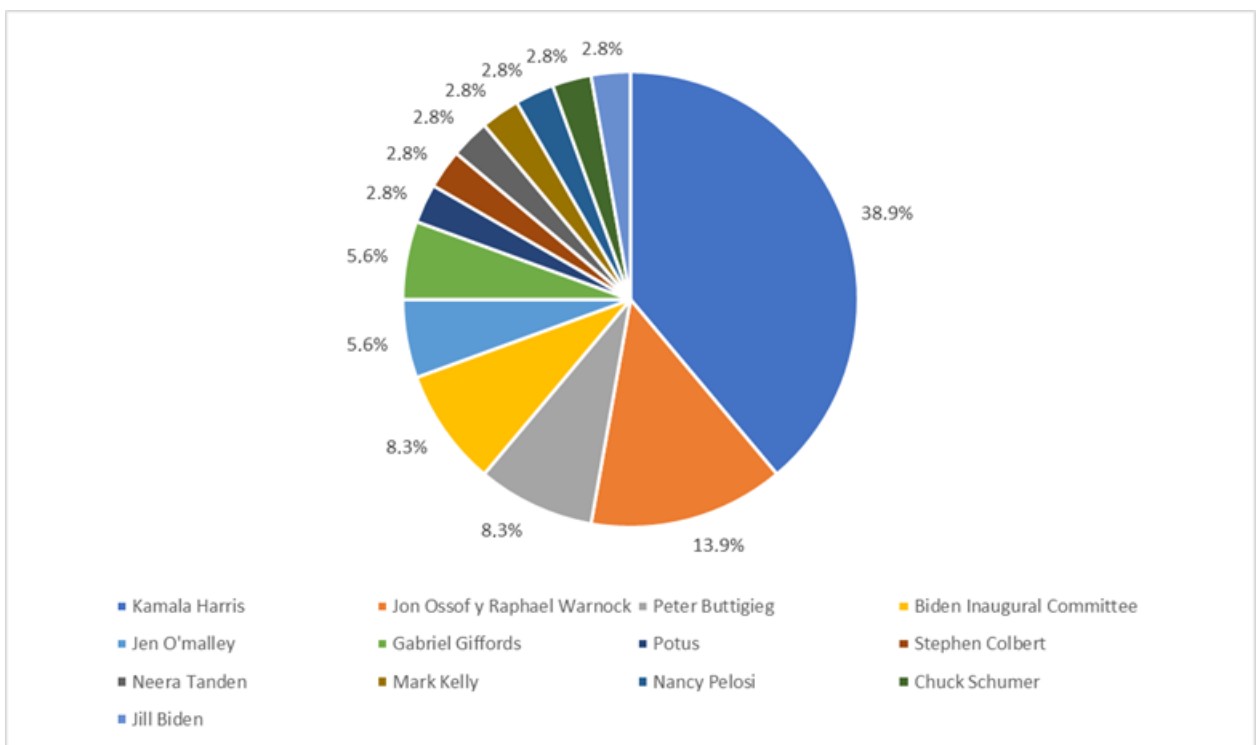

**Figure 5.** Distribution of the accounts cited by Biden.

### 4.4. Keywords for the President-Elect

The analysis of the first 100 keywords of the studied corpus (all of them above a frequency of nine) allowed us to determine the elements that defined Joe Biden's speech after his victory in the 2020 presidential elections.

1. Proper names. The president-elect used many references to the country (America), to the State where the senatorial elections were repeated (Georgia), to himself (Biden), to the future Vice President (Kamala), to her adversary (Trump), and to the future First Lady (Jill). However, there were big differences in the frequency of employment of each word, as Biden often mentioned America (46) or Georgia (45) while outgoing President Donald Trump was mentioned only a few times (14).

2. Appeal to action. Biden's tweets largely added verbs that sought to evidence the ability of his future administration to implement measures that improve people's quality of life (work, build back, care, lead, ensure, heal, relief, and action) or that implied a call to action for citizens (tune in, power, chip in, and join).

3. Building unity. The word most used by Biden was American (all Americans or every American), frequently combined with people (the American people). Furthermore, he usually included references to the family (family), the community (community), the country (nation, country, or United States), and his team (team or group). With these words, the Democrat intended to convey the feeling of belonging to the same country and to promote social cohesion facing Trump's attempts to polarize and divide society.

4. Fight against the pandemic. Another set of words that Joe Biden incorporated in his tweets involved those related to the health crisis situation caused by the COVID-19 pandemic (COVID, lives, pandemic, or virus). The Democrat appealed to the need to face the pandemic (fight, efforts, face to, or head to), such as strengthening the health system (health), vaccination (vaccine), or the use of masks (wear a mask).

5. Political and electoral context. The story was also completed with the requirement that the votes had to be counted (vote must be counted), and calls for an early vote in Georgia (vote early). Other words related to the electoral process also appeared, such as campaign, runoff, or race. On the other hand, Biden made many allusions to his future

administration (Administration, Secretary, or Department) and appeals to the Legislative (Congress or Senate) to promote relief measures for the population.

6. Other political priorities. Throughout his messages, Joe Biden described the main measures that he would promote from the Presidency. The Democrat resorted to thematic keywords that highlighted his political priorities (climate, jobs, justice, plan, business, or science).

7. Defense of his victory. Joe Biden displayed a range of terms related to himself, claiming his victory (win, future, victory, honor, celebrate, or hope). In relation to the word "president", the Democrat respected the official treatment and referred to himself as president-elect and Trump as President Trump.

8. Words of reinforcement. Lastly, he used words that reinforced his speech, including several temporal references (day, today, year, moment, or week).

In short, the presence of keywords in the tweets posted by Joe Biden on Twitter could be related to the issues (issue frame) that featured in his post-election speech. Likewise, the strategies of the president-elect (game frame) on the promotion of political action, the call for unity, and the fight against the pandemic were remarkable. The frequent appearance of words with a positive tone (hope, honor, care, relief, folks, or heal) showed Biden's commitment to flee from the confrontation with Donald Trump.

## 5. Discussion and Conclusions

The US election is a global phenomenon; however, in 2020, its relevance was even larger due to the dissemination of hoaxes and accusations of fraud by Donald Trump, who was candidate and president at the time. In this context, Twitter is an important tool since it works as a key platform for political communication, and Trump used it to post most conspiracies about the elections.

Our study aimed to offer insightful findings on how Biden managed his communication on Twitter as a president-elect. First, Twitter was not used by Biden in a massive way (RQ1). Instead, he provided more messages on selected dates, which matched the milestones of the transition process. Nevertheless, the results illustrated the plan of the Democratic candidate to avoid Trump's aggressive speech and direct confrontation with him. In the context of a global wave of populism [61], this finding demonstrates that the chosen period is relevant, since Biden's tone after knowing the results seems to be presidential.

The second contribution refers to the issues and strategies of Biden on Twitter (RQ2). These two items were key in the current working of political communication, defining messages from objectives. The analysis of issue frames showed a focus on public administration through topics such as the electoral process, COVID-19, economy, or security, but the preference for administration was also observed in game frames. The main strategies were the announcement of measures to fight the crisis caused by the pandemic and the nomination of Cabinet members.

As stated, prior scholarship has consistently described the impact of personalized candidates during electoral contests [62]. Social media works as a political participation equalizer [63] and a place where candidates share messages in line with the proposals in the time of governance. The counter-speech of Biden was not conflictive, instead trying to counteract the theories of conspiracy and fraud from a nonaggressive approach.

In response to RQ3, the most mentioned words (keywords) were also related to issues on public administration that shaped his post-electoral agenda. The tone of the words was mostly positive, and, once again, Trump was ignored. Beyond that, the appeals to action or unity overlapped with the use of game frames provided by Biden. His objective was to promote a political action that moved forward a fragmented society in order to tackle the social challenges. This also explains why Vice President Kamala Harris was cited.

In addition to giving a detailed analysis of the communication action of Biden, who tried to anticipate his future presidential agenda, our research contributes to current discussions on how to interact with polarized audiences. The context is marked by a disinformation order that decreases the citizens' trust of democratic institutions [64]. Our

results evidence that Biden mostly used positive and neutral messages from an institutional approach; however, the public also interact with a negative bias, as shown in the number of retweets and likes. This should promote reflection upon the effects of communicating in a homophilic way on Twitter [65], which turns politicians into elites and distances them from citizenship.

The present research was restricted to the political system and the specific context between the elections and the inauguration of Biden as a president. This time was affected by the massive dissemination of conspiracies, which were relevant to delve into. However, the post-truth area involves journalism as well [66], even considering the visibility of political events on Twitter through media outlets [67]. Accordingly, future studies may expand the scope of this research by considering the relationship of the candidate with the media. It would also be interesting to shed light on the behavior of Biden on Twitter during the electoral transition process and whether it is the same as when he held office.

Moreover, another element to appraise is the platformization of politics [68]. The changes boosted by social platforms could determine the production of political messages. Specifically, the characteristics of Twitter are insightful to understand why Biden chose several issues and strategies on public administration to develop a storytelling on legitimacy. Nevertheless, the huge impact of Trump's digital rhetoric at the international level [69] demands studies that compare the ability of Biden and legacy media to counteract his discourses beyond the US [70].

In conclusion, the empirical findings revealed an original approach of Biden as a president-elect since he faced polarization and the feeling of little political legitimacy through a positive perspective. The population and the role of institutions to solve its problems after a pandemic were placed at the forefront. This pattern was followed not only by the presence of issues and game frames but also by the keywords. His number of tweets was not very high, linked to a typical work schedule. The positive approach and the low level of messages are potential reasons why Biden did not prevent a decline in trust according to polls. Polarization seems to be embedded in American society, arguing that a single president cannot change a structural problem that instead will rely on technological and political participation challenges.

**Author Contributions:** Conceptualization, C.P.-C. and R.D.-G.; methodology, R.R.-d.-R. and R.D.-G.; software, R.D.-G.; validation, R.R.-d.-R.; formal analysis, R.R.-d.-R. and R.D.-G.; investigation, C.P.-C., R.R.-d.-R., and R.D.-G.; resources, R.D.-G.; data curation, R.R.-d.-R.; writing—original draft preparation, R.R.-d.-R.; writing—review and editing, C.P.-C. and R.D.-G.; visualization, R.D.-G.; supervision, C.P.-C.; project administration, C.P.-C. All authors have read and agreed to the published version of the manuscript.

**Funding:** This research received no external funding.

**Data Availability Statement:** Not applicable.

**Conflicts of Interest:** The authors declare no conflict of interest.

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
