# Peer review of "Facing Conspiracies: Biden’s Counter-Speech to Trumpist Messages in the Framework of the 2020 US Elections"

_societies, doi:10.3390/soc12050134_

Round 1

Reviewer 1 Report

Dear authors,

Your manuscript is very interesting and pertinent. I believe it can be a real addition to literature. Still, some issues must be discussed before the manuscript is accepted for publication.

1 - Research objectives and research questions would be better in the introduction section.

2 - Although an in-depth contextualization is carried out, I believe that it is important to know more about the research problem. How can this study be important for the literature? What similar studies exist? What motivated this investigation?

3- It will be important, to improve the manuscript, to refer to the importance of Twitter in political communication or political discourse. Why analyze Biden's political communication on Twitter and not on other social networks?

4 - The authors refer that a content analysis was performed. How many coders performed the analysis? What codebook did you use to analyze the content of Biden's tweets? I suggest you add the codebook you used to the manuscript.

5 - As for the discursive strategies, the authors created several categories of analysis, it would be important to specify what type of tweets make up each of the categories. Also, some text segments of tweets are missing to strengthen the text.

6 - Why did you use this viralization formula?

7 - The authors mention that they used statistical software to analyze the results. What statistical tests did you perform? Where are these statistical tests represented?

Author Response

Response to reviewer 1

Firstly, we would like to thank your comments on the article and its interest as the observations can help to improve the manuscript. In the following paragraphs, we explain how we addressed point-by-point the reviewer’s suggestions:

  1. Research objectives and research questions have been placed at the end of the introduction section to make them visible since the beginning.
  2. The relevance of the research problem is now better justified through this information:

“The literature has consistently described the information disorder and the degree of misinformation, which alerted public institutions [13] and social platforms to intervene [14], sanctioning the electoral use of leaders and content fraudulent of their messages. Nonetheless, it would be relevant to illustrate how political competitors react to disinformation, especially if they develop a counteractive communication. This sort of research contribution could contribute to ongoing works on the battle against distrust in politics”

  1. Twitter has a great relevance in the political discourse of the United States since 2016. This importance is pointed out in the new version:

“Specifically, Twitter was chosen as a social network to study, given its key position for political communication in electoral processes. Likewise, it was the main social platform for Donald Trump in order to disseminate his messages. In 2016, the use of Twitter in the US elections broke all the established distinctions that observers had used to describe the campaigns, starting a new era of political discourse [49]. This makes it relevant to analyze this specific social network”.

  1. The content analysis was performed by three coders. This information, together with the codebook for the agenda, has been included in the materials and methods section.
  2. As required, we mention which type of tweets were analyzed: “our study takes into account individual tweets and responses, but not the retweets, since they are not relevant to find out the Biden’s agenda”. Some text segments have also been added to reinforce the text:

“For instance, on January 4, 2021 Biden posted a message with a positive tone “Georgia — If you haven’t returned your absentee ballot yet, drop it off at a drop box in your county today. If you’re voting in-person tomorrow, make your plan today. Let’s flip the Senate”. By contrast, the neutral or negative tweets were criticism of the conflictive situation “We need to remember: We’re at war with a virus — not with each other” (November 25, 2020)”.

  1. Regarding the viralization formula, we used it because of the need to determine which are the topics that promote a higher engagement. As the new version states, these issues may be useful to reconnect with a distant audience.
  2. Considering statistical tests, we would like to note that SPSS was not employed for statistical tests, but to achieve descriptive findings. This point is now clarified: “The statistical program to process the data was IBM SPSS Statistics, Version 28, which allows to obtain descriptive findings. Statistical tests were not suitable due to the small sample size”.

The writing and the use of English language have been revised to ensure a good comprehension.

We hope we have addressed most of the reviewer’s main concerns.

Sincerely,

The authors

Reviewer 2 Report

The text analyzes Joe Biden's Twitter messages during the period between his election and his inauguration, approximately two and a half months, in which he published 379 messages.

The relationship between some of the issues covered in the title, summary introduction and the research itself is somewhat weak. The contents of Biden's messages and his virality are investigated, but the counter-narrative to Trump or polarization are not delved into, which would require other analytical tools. It would have been interesting, for example, to measure the polarization in the responses or mentions of Biden's messages.

The text contains highly interpretive/evaluative expressions, which are not supported by data or bibliographical references: “Trumpist Illegitimated Messages”, “Joe Biden's era was paradoxically marked by a huge confidence”, “The strategy of the Democratic leader focused on a population devastated by COVID -19 and the information disorder of elections” “the low level of support and the wide list of problems that Biden had to deal with at the beginning of his presidency overlapped with the conservative turn of the US society” “Therefore, the theory that audiovisual messages are more attractive to digital audiences is denied” “This led to an unprecedented attack for a consolidated democracy, which required a public communication to foster the legitimacy of the process”... The formulation of these types of statements should be reviewed or supported by data/studies.

Although the text is based on the analysis of Biden's political communication, much of the introduction is based on Trump's messages. Although it is undoubtedly necessary to explain the context in which Biden's messages are produced, it is also necessary to provide information on the way in which the Democratic Party and Biden approached the campaign informatively. The way in which they approached political communication during the electoral process should have a greater presence on its own, not only in contrast to Trump.

Figure 1 does not help to see the data. Perhaps there would be the possibility of representing them in another way: in two axes, in two graphs, in a table... The rest of the figures are not very clear either.

The following is stated: “With regard to audiences, the analysis of virality of the tweets published by the Democrat (Figure 2) revealed that these messages were very well received by the public on Twitter.” This is difficult to sustain without providing data that allows some kind of comparison (e.g. those of Trump himself, some previous study). In the conclusions it is mentioned: “This finding contrasts a polarized campaign”… The study does not analyze the campaign, but the post-campaign, which could have developed differently. The chosen period is relevant, since Biden's tone after knowing the results seems to be much more "presidential" on the issues. It would be interesting to compare it with some previous data, if any.

The conclusions state “however, the public preferred a negative bias, as shown in the number of retweets and likes”, although this seems to contradict the data (line 259).

In summary, the analysis carried out is interesting, but it would be good to better focus the topic of analysis on the relevant sections of the article and reformulate all the highly evaluative or interpretive statements that are not supported.

Author Response

Thank you for your valuable comments. As suggested, we tried to offer a better focus on the topic; meanwhile, highly interpretative statements have been deleted.

  • We are aware of the relevance of analyzing polarization, but we prefer to further our understanding on Biden’s messages as a way to assess his discourse during a very conflictive time frame.
  • As stated, the highly evaluative expressions cited by the reviewer have been mostly removed.
  • The introduction has been reworked to highlight more the informative role of the Democratic Party and Biden. Specifically, this paragraph was added:

“As the campaign was marked by polarized statements, it is noteworthy to know how the Democratic Party and Biden approached the campaign informatively.

A distrust in the management of public affairs exists together with strongly ideo-logical opposing positions and messages about the illegitimacy of the elections. As long as polarization seems to be a key factor in the shaping of the current US democracy, Biden could develop a consensus approach centered on the idea that “America is back” as collected by international specialized reports [12] that asked for a pact between institutions to respond to citizens' mistrust”.

  • Regarding the figures, we assume that they may not be very clear, but we do think that this is due to the limited template of the journal. A lot of data were managed to create the use of issue frames per day (figure 1); hence, the graphic representation needs to show the daily distribution and probably there is not a better way to do it.
  • The messages of Biden were very well received. To illustrate this, we attach now previous studies on Twitter reception in the 2016 US elections.
  • Moreover, it is also true that the polarization of the campaign was not measured. For this reason, we delete this reference, adding the following information:

“In the context of a global wave of populism [61], this finding demonstrates that the chosen period is relevant, since Biden’s tone after knowing the results seem to be presidential”.

  • There was a mistake concerning the conclusions and the negative bias. The correct statement is the following:

“Our results evidence that Biden mostly used positive and neutral messages from an institutional approach; however, the public also interact with a preferred a negative bias, as shown in the number of retweets and likes”.

The conclusions of the article are highlighted as relevant contributions on political discourses.

We hope we have addressed most of the reviewer’s main concerns.

Sincerely,

The authors

Round 2

Reviewer 1 Report

The manuscript has been accepted for publication.

Author Response

Thank you for the acceptance and your previous comments, which were really helpful to enhance the quality of our contribution. Therefore, the manuscript is ready for publication.

Reviewer 2 Report

Dear authors:

Thank you for revising the manuscript. I think my major concerns have been answered.

A couple of minor things:

On line 43, the new phrase reads as follows: "As the campaign was marked by polarized statements, it is noteworthy to know how the Democratic Party and Biden approached the campaign informatively." The study deals with Biden's tweets once he has been elected president, perhaps it would be better to specify that (as told in line 61).

Perhaps in the editing process some of the resolution of the graphics can be improved, it is a pity that they cannot be seen more clearly.

Best.

Author Response

Dear reviewer,

Thank you for your response. The phrase cited has been reworked to better specify the study. On this matter, the new sentence is "As the campaign was marked by polarized statements, it is noteworthy to know how Biden approached the post-election period on Twitter".

Besides that, the full text has been revised one more time in terms of use of English language.

Regarding the resolution of the graphics, we hope to improve this point in the editing process.

Best,

The authors